Effects of five types of exercise on vascular function in postmenopausal women: a network meta-analysis and systematic review of 32 randomized controlled trials

Sun Weihao
Han Yanli
Gu Song 1275387678@qq.com
Beijing Sport University , Beijing , China
Plavec Davor
Electronic publication date: 2024 Jul 15
Publication date: 2024
Volume: 12
Electronic Location ID: e17621
Received 2024 Apr 12; Accepted 2024 Jun 1
Copyright: © 2024 Sun et al.
Copyright year: 2024
Copyright holder: Sun et al.
License: This is an open access article distributed under the terms of the Creative Commons Attribution License, which permits unrestricted use, distribution, reproduction and adaptation in any medium and for any purpose provided that it is properly attributed. For attribution, the original author(s), title, publication source (PeerJ) and either DOI or URL of the article must be cited.
License URL: https://creativecommons.org/licenses/by/4.0/

Keywords: Vascular function, Vascular structure, Endothelial function, Postmenopausal women, Arterial stiffness, Exercise

Funding: The authors received no funding for this work.

==============================
Background

As women age, especially after menopause, cardiovascular disease (CVD) prevalence rises, posing a significant global health concern. Regular exercise can mitigate CVD risks by improving blood pressure and lipid levels in postmenopausal women. Yet, the optimal exercise modality for enhancing vascular structure and function in this demographic remains uncertain. This study aims to compare five exercise forms to discern the most effective interventions for reducing cardiovascular risk in postmenopausal women.

Methods

The study searched PubMed, Web of Science, Cochrane, EBSCO, and Embase databases. It conducted a network meta-analysis (NMA) of randomized controlled trials (RCTs) on five exercise interventions: continuous endurance training (CET), interval training (INT), resistance training (RT), aerobic combined with resistance training (CT), and hybrid-type training (HYB). Outcome measures included carotid artery intima-media thickness (IMT), nitric oxide (NO), augmentation index (AIx), pulse wave velocity (PWV), and flow-mediated dilatation (FMD) of the brachial artery. Eligible studies were assessed for bias using the Cochrane tool. A frequentist random-effects NMA was employed to rank exercise effects, calculating standardized mean differences (SMDs) with 95% confidence intervals (CIs).

Results

The analysis of 32 studies (n = 1,427) indicates significant increases in FMD with CET, INT, RT, and HYB in postmenopausal women. Reductions in PWV were significant with CET, INT, RT, CT, and HYB. AIx decreased significantly with INT and HYB. CET, INT, and CT significantly increased NO levels. However, no significant reduction in IMT was observed. SUCRA probabilities show INT as most effective for increasing FMD, CT for reducing PWV, INT for decreasing AIx, CT for lowering IMT, and INT for increasing NO in postmenopausal women.

Conclusion

The study demonstrates that CET, INT, RT, and HYB have a significant positive impact on FMD in postmenopausal women. Furthermore, all five forms of exercise significantly enhance PWV in this population. INT and HYB were found to have a significant positive effect on AIx in postmenopausal women, while CET, INT, and CT were found to significantly improve NO levels. For improving vascular function in postmenopausal women, it is recommended to prioritize INT and CT exercise modalities. On the other hand, as CET and RT were not ranked at the top of the Sucra value ranking in this study and were less effective than INT and CT as exercise interventions to improve vascular function in postmenopausal women, it is not recommended that CET and RT be considered the preferred exercise modality.

Introduction

Cardiovascular disease (CVD) which includes high blood pressure, atherosclerosis, heart attack, and stroke, is more prevalent among middle-aged and older adult individuals (Wu et al., 2023). As people age, their organs, systems, and cells gradually deteriorate (Zeng et al., 2022), which is particularly evident in the cardiovascular system. The aging process primarily results in myocardial remodeling (Rose & Howlett, 2024), declining cardiac function (Ali et al., 2020), reduced functional reserves of the heart and blood vessels (Zhang et al., 2024), arteriosclerosis (Shikimoto et al., 2024), and increased endothelial damage (Li, 2019). Additionally, metabolic changes, long-term unhealthy dietary habits, lack of exercise (Panzarino et al., 2017), and various other factors are associated with this process, which cumulatively contribute to the onset of CVD in the older adult population. CVD is strongly associated with mortality, accounting for almost 30% of all deaths in middle-aged and older adult population (Coutinho et al., 2023). In certain countries, this percentage is even higher, exceeding 50% (Brink et al., 2017). Therefore, it is one of the most significant health concerns affecting the older adult population today.

Postmenopausal women, are at a higher risk of CVD due to a significant decline in estrogen levels and the physiological degradation of ovarian function (Fischer & Haffner-Luntzer, 2022). Estrogen is essential for vasodilation, maintaining healthy lipid levels, suppressing inflammation, and exerting vital protective and regulatory effects on the cardiovascular system (Iorga et al., 2017). Estrogen deficiency in postmenopausal women may lead to lipid metabolism disorders, decreased vascular elasticity, and endothelial dysfunction, thereby increasing the risk of atherosclerosis, hypertension, arteriosclerosis, and other CVD (Aryan et al., 2020). Epidemiological studies have shown that the prevalence of cardiovascular disease in premenopausal women is lower than in men. However, after menopause, the risk of cardiovascular disease significantly increases, even surpassing that of men of the same age (Gersh et al., 2024; Teoh et al., 2020). According to research conducted by the American Heart Association, the occurrence of cardiovascular events in postmenopausal women is two to three times higher than in premenopausal women (Rodgers et al., 2019). Therefore, it is crucial to address cardiovascular health issues in postmenopausal women.

Estrogen replacement therapy is a common clinical treatment for preventing and improving the occurrence of CVD in postmenopausal women. However, this method carries potential risks, such as weight gain, abnormal uterine bleeding, breast discomfort, and an increased risk of breast cancer (Mattix & Singh, 2000; Speth et al., 2018). Therefore, physical activity has become a highly recommended non-pharmacological treatment for CVD (Haring et al., 2023). Regular physical exercise is widely recognized as an effective intervention for reducing the risk of CVD and is free from potential side effects (LaMonte et al., 2018). Research suggests that physical exercise can improve lipid metabolism (Yin et al., 2023), endothelial function, vascular elasticity (Dai, Rabinovitch & Ungvari, 2012), and positively impact vascular health (Seals, Nagy & Moreau, 2019). Regular exercise that meets or exceeds the recommendations of physical activity guidelines is significantly associated with a reduced risk of CVD and mortality (Tucker et al., 2022). Expending at least 1,000 kcal of energy through exercise per week can reduce CVD mortality by 20% (Kokkinos, 2012). However, further exploration is required to determine the specific exercise modalities. Most studies and cardiovascular exercise guidelines commonly recommend endurance training (CET) or aerobic combined with resistance training (CT) as the preferred methods for improving vascular health in postmenopausal women (Lee et al., 2024; Pelliccia et al., 2020). These methods have yielded significant results, although they require a substantial time commitment. According to Atakan et al. (2021)’s findings, interval training (INT) is more effective and time-efficient in improving lipid metabolism compared to continuous endurance training (CET). However, it poses some cardiovascular risks for postmenopausal women due to its higher intensity (Lee et al., 2019). The benefits of resistance training (RT) on vascular health are still a matter of debate, with the American Heart Association suggesting positive effects on patients with CVD (Williams et al., 2007). However, according to Schroeder et al. (2019), there was no significant improvement in blood pressure after 8 weeks of resistance training (RT). Furthermore, other training modalities, such as hybrid-type training (HYB), have gained increasing attention. Hybrid-type training (HYB), which consists of various components, can exert different intensity training effects on the cardiovascular system during a single training session (Batrakoulis et al., 2019b, 2019a). Batrakoulis et al. (2021), Sperlich et al. (2017), Feito et al. (2019), Batrakoulis et al. (2018) and other studies have shown that hybrid-type training (HYB) can effectively improve body composition, cardiorespiratory function, and the cardiovascular system in female populations. However, hybrid-type training (HYB) can be complex and challenging, especially for the older adult population (Batrakoulis et al., 2023). Therefore, further exploration is needed to investigate the effects of different exercise modalities on vascular health in postmenopausal women. To improve vascular health in postmenopausal women while minimizing potential risks, it is necessary to have clearer evidence when selecting the most suitable exercise modality.

Currently, most meta-analyses that examine the effects of exercise on vascular health in postmenopausal women focus on single types of exercise and limited outcome measures (Brislane et al., 2022; Saquetto et al., 2022). These analyses only reveal the health benefits of a particular type of exercise on blood pressure and lipid levels (Xin et al., 2022; Zhou et al., 2023; Debray et al., 2023), with fewer studies comparing the effects of different types of exercise on vascular health in postmenopausal women. It is important to note that these findings are limited to the specific outcome measures studied and do not provide a comprehensive understanding of the effects of exercise on overall vascular health in this population. In addition to comparing exercise types, there is a lack of research that uses comprehensive indicators to evaluate and compare vascular health in postmenopausal women. These indicators include endothelial function, arterial stiffness, and key biomarkers such as vascular endothelial factors (Xi et al., 2021).

This study utilizes a network meta-analysis (NMA) approach to compare and analyze the effects of different forms of exercise on vascular health in postmenopausal women. The aim is to explore the most effective type of exercise and clarify which exercise training modality can maximally reduce cardiovascular risk in postmenopausal women. The impact of different types of exercise on various indicators of vascular health in postmenopausal women, such as carotid artery intima-media thickness (IMT), nitric oxide (NO), augmentation index (AIx), pulse wave velocity (PWV), and flow-mediated dilatation (FMD), is investigated in this study. This study is the first attempt to systematically evaluate the effects of different types of exercise on vascular health in postmenopausal women. The study covers vascular structure, arterial dilatation function, arterial stiffness, and endothelial function. The results provide scientific evidence for exercise prescriptions and health management in postmenopausal women.

Methods

Registration

The literature search date for this study was January 1, 2000–December 31, 2023, and our actual literature search for this study was January 5, 2024.

The research protocol for this NMA was registered in the International Prospective Register of Systematic Reviews (identifier: CRD42024498890) and strictly adheres to the PRISMA (Preferred Reporting Items for Systematic Reviews and Meta-Analyses) guidelines for NMA (Hutton et al., 2015).

Literature search strategy

We conducted a systematic search in several databases including PubMed, Web of Science, Cochrane, EBSCO, and Embase. We applied search terms using MeSH terms in PubMed/Cochrane and Emtree terms in Embase. The primary terms used to retrieve relevant literature included exercise, training, aerobic exercise, moderate intensity continuous training, resistance training, combined training, and high-intensity interval training, combined with postmenopausal, postmenopausal*, older adult women, vascular function, endothelium, pulse wave velocity, and flow-mediated dilation. The search strategy was based on key phrases relevant to the PICOS (Population, Intervention, Comparison, Outcome, Study Design) tool: (P) Population: postmenopausal women and/or older adult women; (I) Intervention: CET, INT, RT, CT, HYB, and CON (as listed in Table 1); (C) Comparison: no exercise; (O) Outcome: FMD, PWV, AIx, IMT, NO; (S) Study type: RCTs. Please refer to Appendix S2 for details of the search strategy.

Table 1 Definition of exercise types.

Type	Definition	
CET	Intensity: >85% VO2max or >85% HRR or >85% HRmax	
Type: any continuous traditional aerobic exercise mode (such as walking, running, cycling, rowing, swimming, circuit training, moderate-intensity continuous training, and step exercises)	
INT	Intensity: >85% VO2max or >85% HRR or >85% HRmax	
Type: any intermittent traditional interval training mode (such as walking, running, cycling, rowing, swimming, elliptical training, moderate-intensity interval training, sprint interval training, high-intensity interval training)	
RT	Intensity: >65% 1 RM or >100% 1 RM	
Type: any mode of resistance training, including resistance training of body muscles (such as free weights, weight machines, resistance bands)	
CT	A combination of CET and RT	
HYB	Intensity: >85% VO2max or >85% HRR or >85% HRmax	
Type: any comprehensive multi-component exercise mode other than CET, RT, CT, or INT, involving simultaneous training of cardiovascular and musculoskeletal systems at different intensities during exercise (such as WBV training, exergaming, taekwondo, integrated neuromuscular training, cardiac resistance training, and mind-body exercises).	
CON	Type: no exercise or light stretching	
Note:

CON, control; CET, continuous endurance training; INT, interval training; RT, resistance training; CT, combined training; HYB, hybrid-type training; WBV training, whole-body vibration training; VO2max, maximal oxygen uptake; HRmax, maximum heart rate; HRR, heart rate reserve.

Eligibility criteria

The study was deemed eligible if it satisfied the following criteria: (1) it involved postmenopausal or older female participants; (2) it included interventions such as CET, INT, RT, CT, or HYB (as listed in Table 1); (3) it evaluated at least one vascular function indicator: FMD, PWV, AIx, IMT, or NO; (4) the experimental group underwent structured exercise training for ≥6 weeks; (5) it encompassed all English-language RCTs published from 2000 to December 2023.

Studies were excluded if they: (1) were duplicates, literature reviews, editorials, letters to editors, conference abstracts, or animal studies; (2) full text was unavailable or complete study data was difficult to obtain; (3) did not report on the vascular function indicators of interest for this review; (4) involved mixed interventions with exercise and other measures, e.g., exercise combined with medication, diet, etc.

Study selection

Title and abstract screening, as well as full-text screening, were conducted independently by two authors (Weihao Sun, Yanli Han) in accordance with the inclusion and exclusion criteria. Any disputes were resolved by Song Gu. The references of the selected articles were manually searched to identify articles that met the inclusion criteria.

Data extraction

Data from the included studies were independently extracted by two authors, Weihao Sun and Yanli Han. Any disagreement was resolved by the opinion of Song Gu. The literature provided the following information: the first author, publication year, country, and demographic characteristics of the participants (including the number of participants in the experimental and control groups, age, and comorbidities), details of the exercise intervention (such as the type of exercise, intensity, duration, frequency, session length, and whether it was supervised or unsupervised), and relevant outcome measures. If data was unavailable in the articles, we attempted to contact the authors to obtain missing information. Please refer to for further details.

Risk of bias assessment

The risk of bias (ROB) of the included studies was assessed by two researchers, Weihao Sun and Yanli Han, using the Cochrane Risk of Bias Assessment Tool (Nasser, 2020). Due to the nature of exercise interventions, blinding of participants was not feasible in the included studies (Huang et al., 2021). Therefore, we only assessed the other six domains of bias, which include random sequence generation, allocation concealment, blinding of outcome assessment, incomplete outcome data, selective reporting, and other sources of bias. We assessed the overall ROB for each study as follows: (1) Low ROB: studies not rated as having high ROB and with ≤3 domains judged as unclear risk (Chen et al., 2024); (2) Moderate ROB: studies with ≤1 domain rated as high ROB, but with ≥4 domains judged as unclear risk; (3) High ROB: all other scenarios categorized as high ROB (Cipriani et al., 2018).

Data synthesis and statistical analysis

This NMA estimated the effect by combining the pre-to-post changes in the experimental and control groups. SMDs and their 95% confidence intervals (Cis) for FMD, PWV, AIx, IMT, and NO were calculated. Random-effects multivariate NMA was conducted within a frequency framework (Salanti, 2012; Bucher et al., 1997) using STATA 17.0 software (StataCorp, College Station, TX, USA), following the current PRISMA NMA guidelines (Hutton et al., 2015). Estimates of combined effects, along with their 95% CIs and 95% prediction intervals (PrI), are presented.

A network diagram is generated to visually display the relationships between different intervention measures. The lines connecting the nodes represent direct comparisons between intervention measures, with the size of each node and the thickness of the connecting lines proportional to the number of studies. In addition, a network contribution plot is created to calculate the contributions of each direct comparison. The inclusion criteria of each study are assessed to determine whether postmenopausal women can be randomly assigned to receive any intervention. Consistency models are then used to evaluate the assumption of transitivity (Rücker & Schwarzer, 2015). Transitivity is a fundamental assumption of NMA. NMA assumes that indirect comparisons offer valid estimates of unobserved direct comparisons (Salanti, 2012). It also assumes that the effect modifiers of all studies follow a uniform distribution (Jansen & Naci, 2013). To assess the consistency of each closed loop, we calculate inconsistency factors (IFs) and their 95% CIs. We check for inconsistency using inconsistency models. When there is no significant inconsistency, a consistency model is adopted (P > 0.05) (Shim et al., 2017). Node-splitting analysis is conducted to examine local inconsistencies, ensuring reliable results (P > 0.05).

The surface under the cumulative ranking curve (SUCRA) was used to rank and compare the effects of different types of exercise interventions (Salanti, Ades & Ioannidis, 2011). SUCRA values range from 0 to 100, with higher values indicating better effects of exercise intervention (Mbuagbaw et al., 2017). Furthermore, a network funnel plot is constructed to test for the presence of publication bias in NMA, and a symmetry criterion is used for intuitive examination.

Statistical processing

Statistical processing data were analyzed by META through Stata17.0 software, with outcome indicators as continuous variables. Due to the varying use of tools for the same indicator in the included literature, the standardized mean difference (SMD) was used to indicate the baseline, and the baseline was uniformly adjusted to α = 0.05. Heterogeneity was tested using the Q-test and I2-test, and publication bias analysis was performed using the Egger’s test. In reticulated meta-analysis, if there was a closed-loop structure in the reticulated evidence graph, the inconsistency test was performed using nodal analysis. The results of the ring inconsistency test were analyzed for consistency modeling if P > 0.05. Local inconsistency tests were also conducted using node splitting, and if there was a result of P < 0.05, the results were directly compared in the conventional META analysis following this comparison. The results were analyzed by the SUCRA value of the area under the cumulative ranking probability map (surface under the cumulative ranking, SUCRA) to rank the superiority of each intervention (0 ≤ SUCRA ≤ 1).

Results

Literature selection

Figure 1 depicts the flowchart of the literature search and screening process. The search strategy identified 645 articles, 643 from databases and two from reference lists. After removing 179 duplicates, 466 articles remained for screening. Of these, 298 were excluded after screening titles and abstracts. Upon full-text review, 136 articles were further excluded, leaving 32 articles eligible for quantitative synthesis.

Figure 1 Flow chart of literature screening.

Literature characteristics

The characteristics of the studies are outlined in Appendix S3, and the list of studies is provided in Appendix S4. The studies were published between 2000 and December 2023. The experimental group comprised 866 participants (CET: 341; RT: 235; INT: 83; CT: 68; HYB: 139), while the control group consisted of 561 participants aged between 50 and 70 years. Fourteen studies included participants with symptoms of obesity, hypertension, and cardiovascular risk. One study (Kujawski et al., 2018) included two males in each of the control group (n = 28) and experimental group (n = 27). Several included studies did not report the intensity of the HYB group, and the intensity of INT studies varied widely (e.g., 55–85% HRmax). The duration of exercise listed in Appendix S3 does not include warm-up, cool-down, or stretching time. Exercise interventions lasted for ≥6 weeks, with over half (78.12%) of the studies conducting interventions for 12 weeks. Twenty-three studies provided supervised training, six studies combined supervised and remote training, and four studies used remote training alone. The literature analyzed in this study mainly focused on FMD, PWV, AIx, IMT, and NO. Therefore, this NMA analyzed FMD, PWV, AIx, IMT, and NO. Although other vascular biomarkers, such as Endothelin-1 (ET-1) and β-stiffness index, were also studied, the data on these vascular biomarkers were insufficient to conduct a NMA.

Risk of bias assessment results

The risk of bias assessment results for each study are detailed in Appendix S5. Of the included studies, 30 reported clear randomization methods, 27 reported allocation concealment, and 23 described blinding in outcome assessment. A dropout rate exceeding 20% was considered high risk, and three studies were classified as high risk due to this criterion (Chen et al., 2024). Furthermore, 26 studies showed low risk of selective reporting. Among other biases, two studies were rated as high risk due to significant measurement errors in outcome assessment. In summary, 24 studies were judged to have low ROB, six had moderate ROB, and two had high ROB.

Network meta-analysis

Figure 2 shows the network diagram of eligible studies investigating the effectiveness of different exercise modalities on all five vascular function indicators. The size of the nodes corresponds to the sample size of each exercise modality, while the thickness of the lines between exercise modalities reflects the number of studies involved in that comparison. The most common intervention type is CET, while INT and CT have the lowest frequency. The contribution plot for network comparisons, both direct and indirect, and the number of studies for each direct comparison can be found in Appendix S6. Figure 3 shows forest plots of eligible comparisons of FMD, PWV, AIx, IMT, NO.

Figure 2 Reticular evidence diagram.

Figure 3 Forest plots of eligible comparisons of flow mediated dilation (FMD), pulse wave velocity (PWV), augmentation index (AIx), intima-media thickness (IMT), nitric oxide (NO).

Note: CON, control; CET, continuous endurance training; CT, combined training; HYB, hybrid-type training; INT, interval training; RT, resistance training.

The study evaluated the inconsistency of vascular function outcome indicators through heterogeneity estimates, inconsistency models, and node-splitting analyses (refer to Appendix S7). Heterogeneity estimates for each loop showed non-significant consistency in FMD, PWV, IMT, NO, and AIx (P > 0.05), allowing for further comparisons. The inconsistency model demonstrated non-significance in both direct and indirect comparisons (P > 0.05). Node-splitting analysis showed no significant difference in both direct and indirect evidence (P > 0.05), indicating reliable results. Appendix S8 presents forest plots comparing qualified outcomes of vascular function indicators, including 95% CI and 95% PrI. Appendix S9 provides funnel plots for vascular function outcome indicators, which were used to examine NMA publication bias and small sample effect estimates. The funnel plots of included studies suggest a low likelihood of publication bias or small sample effects in this NMA. Appendix S10 lists the SUCRA probabilities for each intervention on vascular function outcomes, with higher SUCRA values indicating a higher probability of ranking the exercise intervention favorably in the network.

FMD results

Ten studies assessed FMD (N = 394). The results show that compared to the control group, CET (0.91 [0.44, 1.39], P < 0.01), INT (2.66 [1.41, 3.91], P < 0.01), RT (1.09 [0.19, 1.99], P < 0.05), and HYB (1.00 [0.41, 1.60], P < 0.01) all demonstrated significant improvement in FMD in postmenopausal women (refer to Table 2). Table 3 and Appendix S10 demonstrate that all interventions, including INT (SUCRA = 99.30), RT (SUCRA = 55.70), HYB (SUCRA = 51.00), and CET (SUCRA = 43.80), were more effective than the control group (SUCRA = 0.30) in improving FMD in postmenopausal women.

Table 2 Network meta-analysis matrix of results.

FMD	
CET	1.75 (0.50, 2.99)	0.17 (−0.72, 1.07)	0.09 (−0.60, 0.77)	−0.91 (−1.39, −0.44)		
−1.75 (−2.99, −0.50)	INT	−1.57 (−3.07, −0.07)	−1.66 (−3.02, −0.30)	−2.66 (−3.91, −1.41)		
−0.17 (−1.07, 0.72)	1.57 (0.07, 3.07)	RT	−0.09 (−1.03, 0.85)	−1.09 (−1.99, −0.19)		
−0.09 (−0.77, 0.60)	1.66 (0.30, 3.02)	0.09 (−0.85, 1.03)	HYB	−1.00 (−1.60, −0.41)		
0.91 (0.44, 1.39)	2.66 (1.41, 3.91)	1.09 (0.19, 1.99)	1.00 (0.41, 1.60)	CON		
PWV	
CET	−0.31 (−0.90, 0.27)	0.07 (−0.44, 0.58)	−0.71 (−1.50, 0.08)	−0.19 (−0.71, 0.32)	0.48 (0.12, 0.85)	
0.31 (−0.27, 0.90)	INT	0.38 (−0.22, 0.98)	−0.40 (−1.23, 0.43)	0.12 (−0.48, 0.71)	0.79 (0.33, 1.26)	
−0.07 (−0.58, 0.44)	−0.38 (−0.98, 0.22)	RT	−0.78 (−1.58, 0.02)	−0.27 (−0.74, 0.21)	0.41 (0.05, 0.77)	
0.71 (−0.08, 1.50)	0.40 (−0.43, 1.23)	0.78 (−0.02, 1.58)	CT	0.52 (−0.28, 1.31)	1.19 (0.49, 1.89)	
0.19 (−0.32, 0.71)	−0.12 (−0.71, 0.48)	0.27 (−0.21, 0.74)	−0.52 (−1.31, 0.28)	HYB	0.68 (0.31, 1.05)	
−0.48 (−0.85, −0.12)	−0.79 (−1.26, −0.33)	−0.41 (−0.77, −0.05)	−1.19 (−1.89, −0.49)	−0.68 (−1.05, −0.31)	CON	
AIx	
CET	−0.46 (−1.52, 0.60)	0.72 (−0.49, 1.93)	−0.20 (−1.24, 0.83)	0.31 (−0.61, 1.23)		
0.46 (−0.60, 1.52)	INT	1.18 (0.23, 2.12)	0.25 (−0.46, 0.97)	0.77 (0.24, 1.30)		
−0.72 (−1.93, 0.49)	−1.18 (−2.12, −0.23)	RT	−0.92 (−1.68, −0.16)	−0.41 (−1.19, 0.38)		
0.20 (−0.83, 1.24)	−0.25 (−0.97, 0.46)	0.92 (0.16, 1.68)	HYB	0.52 (0.03, 1.00)		
−0.31 (−1.23, 0.61)	−0.77 (−1.30, −0.24)	0.41 (−0.38, 1.19)	−0.52 (−1.00, −0.03)	CON		
IMT	
CET	0.22 (−0.06, 0.50)	−0.14 (−0.63, 0.35)	0.14 (−0.55, 0.83)	0.14 (−0.13, 0.41)		
−0.22 (−0.50, 0.06)	RT	−0.36 (−0.86, 0.15)	−0.08 (−0.78, 0.62)	−0.08 (−0.37, 0.21)		
0.14 (−0.35, 0.63)	0.36 (−0.15, 0.86)	CT	0.28 (−0.48, 1.04)	0.28 (−0.13, 0.69)		
−0.14 (−0.83, 0.55)	0.08 (−0.62, 0.78)	−0.28 (−1.04, 0.48)	HYB	0.00 (−0.64, 0.64)		
−0.14 (−0.41, 0.13)	0.08 (−0.21, 0.37)	−0.28 (−0.69, 0.13)	−0.00 (−0.64, 0.64)	CON		
NO	
CET	4.16 (2.66, 5.66)	0.00 (−2.03, 2.04)	2.57 (0.13, 5.00)	−1.65 (−2.62, −0.67)		
−4.16 (−5.66, −2.66)	INT	−4.16 (−6.42, −1.89)	−1.60 (−4.22, 1.03)	−5.81 (−7.20, −4.42)		
−0.00 (−2.04, 2.03)	4.16 (1.89, 6.42)	RT	2.56 (−0.29, 5.42)	−1.65 (−3.44, 0.14)		
−2.57 (−5.00, −0.13)	1.60 (−1.03, 4.22)	−2.56 (−5.42, 0.29)	CT	−4.21 (−6.44, −1.98)		
1.65 (0.67, 2.62)	5.81 (4.42, 7.20)	1.65 (−0.14, 3.44)	4.21 (1.98, 6.44)	CON		
Note:

Effects are expressed as effect sizes (95% CI) between interventions. Bold indicates a significant effect of the exercise intervention. Abbreviations: CON, control, CET, continuous endurance training; CT, combined training; HYB, hybrid-type training; INT, interval training; RT, resistance training.

Table 3 Ranking of exercise interventions in order of effectiveness.

FMD	PWV	AIx	IMT	NO	
Interventions	SUCRA
(%)	Interventions	SUCRA
(%)	Interventions	SUCRA
(%)	Interventions	SUCRA
(%)	Interventions	SUCRA
(%)	
INT	99.30	CT	93.00	INT	85.70	CT	81.90	INT	90.60	
RT	55.70	INT	71.50	HYB	75.10	CET	68.70	CT	82.70	
HYB	51.00	HYB	61.70	CET	48.80	HYB	41.70	RT	43.20	
CET	43.80	CET	40.40	RT	20.70	CON	35.90	CET	29.10	
CON	0.30	RT	32.90	CON	19.80	RT	21.80	CON	4.50	
CON	0.40	
Note:

Abbreviations: CON, control, CET, continuous endurance training; CT, combined training; HYB, hybrid-type training; INT, interval training; RT, resistance training.

PWV results

Eighteen studies assessed PWV (N = 644). The results show that postmenopausal women who underwent CET (−0.48 [−0.85, −0.12], P < 0.05), INT (−0.79 [−1.26, −0.33], P < 0.01), RT (−0.41 [−0.77, −0.05], P < 0. 05), CT (−1.19 [−1.89, −0.49], P < 0.01), or HYB (−0.68 [−1.05, −0.31], P < 0.01) had a significant improvement in PWV compared to the control group (Table 2, N = 644). Table 3 and Appendix S10 demonstrate that all interventions, including CT (SUCRA = 93.00), INT (SUCRA = 71.50), HYB (SUCRA = 61.70), CET (SUCRA = 40.40), and RT (SUCRA = 32.90), were more effective than the control group (SUCRA = 0.40) in improving PWV in postmenopausal women.

AIx results

Five studies evaluated AIx (N = 163). The results show that compared to the control group, INT (−0.77 [−1.30, −0.24], P < 0.01) and HYB (−0.52 [−1.00, −0.03], P < 0.05) demonstrated significant improvement in AIx in postmenopausal women (refer to Table 2). Table 3 and Appendix S10 demonstrate that all interventions, including INT (SUCRA = 85.70), HYB (SUCRA = 75.10), CET (SUCRA = 48.80), and RT (SUCRA = 20.70), were more effective than the control group (SUCRA = 19.80) in improving AIx in postmenopausal women.

IMT results

Five studies evaluated IMT (N = 415). The network comparison did not show a significant reduction in IMT (refer to Table 2). The effectiveness ranking of different exercise modalities for improving IMT in postmenopausal women was as follows: CT (SUCRA = 81.90), CET (SUCRA = 68.70), HYB (SUCRA = 41.70). All intervention measures, except for RT (SUCRA = 21.80), were superior to the control group (SUCRA = 35.90), as shown in Table 3, Appendix S10.

NO results

Six studies evaluated the effect of NO in postmenopausal women (N = 177). The results indicate that CET (1.65 [0.67, 2.62], P < 0.01), INT (5.81 [4.42, 7.20], P < 0.01), and CT (4.21 [1.98, 6.44], P < 0.01) significantly improved NO compared to the control group (Table 2). The effectiveness ranking of different exercise modalities for improving NO in postmenopausal women was INT (SUCRA = 90.60), CT (SUCRA = 82.70), RT (SUCRA = 43.20), and CET (SUCRA = 29.10). All interventions outperformed the control group (SUCRA = 4.50), as shown in Table 3 and Appendix S10.

Discussion

The systematic review comprised 32 RCTs involving 1,427 postmenopausal female participants. The RCTs examined the effects of five different exercise modalities on FMD, PWV, AIx, IMT, and NO. The NMA results indicated that CET, INT, RT, and HYB significantly improved FMD in postmenopausal women. All five forms of exercise can significantly improve PWV in postmenopausal women. INT and HYB significantly improved AIx in postmenopausal women, while CET, INT, and CT significantly improved NO levels. This study did not find any significant effects of the five exercise modalities on IMT in postmenopausal women. Based on the SUCRA probability ranking results, INT and CT are the exercise interventions with the highest probability of being the most effective for improving FMD, PWV, AIx, IMT, and NO in postmenopausal women.

FMD can serve as a non-invasive indicator of endothelial function in the human body. Endothelial dysfunction is one of the predictive factors for early CVD (Celermajer et al., 1994). Among the five different types of exercise, CET (0.91 [0.44, 1.39]), INT (2.66 [1.41, 3.91]), RT (1.09 [0.19, 1.99]), and HYB (1.00 [0.41, 1.60]) significantly increase FMD in postmenopausal women. This increase in FMD may potentially lower the risk of arterial stiffening and cardiovascular events in postmenopausal women (Lyall et al., 2022). The improvement of FMD in postmenopausal women with CET aligns with the findings of meta-analyses by Ashor et al. (2015), Early et al. (2017), and others. Similarly, INT significantly improves FMD in postmenopausal women. He et al. (2022) conducted an 8-week high-intensity interval training study with 18 postmenopausal women, demonstrating a significant increase in FMD with high-intensity interval training. The study’s SUCRA value ranking indicates that INT is the most effective exercise modality for increasing FMD in postmenopausal women. Furthermore, Jaime et al. (2019) suggested that RT also significantly improves FMD in postmenopausal women, while Fetter et al. (2020) demonstrated that 12 weeks of yoga training significantly improved FMD. The study discovered that exercise led to improvements in FMD for postmenopausal women with mild to moderate depression (Prakhinkit et al., 2014), cardiovascular risk (Jo et al., 2020), and hypertension (Azadpour, Tartibian & Koşar, 2017; Fetter et al., 2020; Swift et al., 2012). This indicates that postmenopausal women with comorbidities can enhance their FMD through exercise (Brislane et al., 2022; Lew, Ethier & Pyke, 2022), which is in line with Lyall et al. (2022)’s findings.

PWV elevation is a significant risk factor for atherosclerosis, hypertension, and heart disease (Zaydun et al., 2006; Han et al., 2013). In postmenopausal women, CET (−0.48 [−0.85, −0.12]), INT (−0.79 [−1.26, −0.33]), RT (−0.41 [−0.77, −0.05]), CT (−1.19 [−1.89, −0.49]), and HYB (−0.68 [−1.05, −0.31]) significantly reduce PWV, improving vascular wall structure and function, and reducing arterial stiffness and plaque formation (Figueroa et al., 2014). INT has been shown to significantly improve PWV in postmenopausal women, which is consistent with the findings of Ho et al. (2020), Wong et al. (2018). The improvement in PWV due to training may be attributed to the reduction of ET-1 (Maeda et al., 2015), which leads to decreased vascular resistance and tension in peripheral arteries (Park et al., 2016). Additionally, exercise-induced increase in shear stress may improve vascular dilation, thereby reducing arterial stiffness (Duprez, 2010). Furthermore, the NMA findings suggest that CET, RT, CT, and HYB are effective in significantly reducing PWV in postmenopausal women, which is consistent with the results of several previous studies (Park et al., 2023; Ohta et al., 2012; Miura et al., 2015; Figueroa et al., 2013; Son et al., 2017; Lee et al., 2019).

AIx is often used as an indicator to assess arterial stiffness and cardiovascular health. It reflects the elasticity and diastolic performance of arteries (Montero, Roche & Martinez-Rodriguez, 2014). In postmenopausal women, INT (−0.77 [−1.30, −0.24]) and HYB (−0.52 [−1.00, −0.03]) significantly decrease AIx, thereby improving arterial stiffness and promoting cardiovascular health (Figueroa et al., 2014). INT can significantly reduce AIx in postmenopausal women. Ho et al. (2020) conducted an 8-week interval training intervention with 30 postmenopausal women, which demonstrated a significant decrease in AIx. Additionally, HYB has been shown to significantly reduce AIx in postmenopausal women, consistent with the findings of Jaime et al. (2019) and Wong et al. (2016).

IMT is used to assess vascular structure and atherosclerosis (Nezu et al., 2016; Crouse et al., 2007). This NMA did not find significant improvements in IMT outcomes in postmenopausal women across the five types of exercise. However, SUCRA values suggest that CT, CET, and HYB may be potential exercise modalities for improving IMT outcomes in postmenopausal women. Tanahashi et al. (2014) and Akazawa et al. (2013) suggest that aerobic training for 8–12 weeks may not significantly improve IMT outcomes in postmenopausal women, but long-term training may have positive effects. For example, one study found a reduction in IMT after 24 weeks of aerobic exercise training (Spence et al., 2013), indicating that changes in IMT may be influenced by the duration of exercise intervention. Park & Park (2017) study showed significant improvements in IMT outcomes in postmenopausal women after 6 months of CT. This finding is supported by the SUCRA ranking in the study (Table 3). Future research could explore the effects of other exercise interventions, such as INT and HYB, on IMT in postmenopausal women.

NO plays a crucial role in vascular dilation (Bourque, Davidge & Adams, 2011) and reducing arterial stiffness (Wong et al., 2018). Among the five exercise modalities, CET (1.65 [0.67, 2.62]), INT (5.81 [4.42, 7.20]), and CT (4.21 [1.98, 6.44]) significantly increase NO levels in postmenopausal women. The increase in blood flow and shear force after exercise intervention may be the reason for the reduction in the inhibition and degradation of NO by free radicals (Belardinelli et al., 2005), as well as the enhancement of vascular elasticity and function (Di Francescomarino et al., 2009). In a 12-week study on aerobic exercise intervention (Prakhinkit et al., 2014), NO levels were significantly increased in postmenopausal women. In a separate study on interval exercise intervention, HIIT significantly increased NO levels in postmenopausal women (He et al., 2022). Additionally, research has shown that combining aerobic and RT also significantly increases NO levels in postmenopausal women (Son et al., 2017), which is consistent with the findings of this NMA.

Statistical analysis was also attempted on other vascular function markers, such as ET-1 and β-stiffness, which have been found to increase arterial stiffness and impair endothelial function (Moreau et al., 2003; Tanaka, Desouza & Seals, 1998). However, limited research has been conducted on these vascular function markers, indicating that they are still in the preliminary exploration stage. Future studies should aim to explore the relationship between vascular function markers and exercise, as well as vascular health, in postmenopausal women. This will contribute to a better understanding of vascular health and cardiovascular disease mechanisms. Additionally, it will provide more exercise strategies for prevention and treatment.

Strengths and limitations

The study has several strengths. Firstly, it utilized five types of exercise modalities that are easily accessible, widely applicable, and cost-effective. This provides motivation and convenience for postmenopausal women to engage in physical activity. Secondly, this study is the first to use NMA to assess the impact of various exercise modalities on five vascular function indicators in postmenopausal women. The study assesses the impact on vascular health by examining vascular structure, arterial endothelial function, arterial stiffness, and endothelial function. This analysis provides scientific evidence for exercise prescriptions and cardiovascular health management in postmenopausal women, helping to develop personalized and effective exercise regimens.

Limitations of this study include its focus on literature published only between 2000 and 2023 and its restriction to English-language publications. To enhance comprehensiveness, future research should expand the scope of literature inclusion.

Conclusion

The presented NMA provides evidence that CET, INT, RT and HYB significantly improve FMD in postmenopausal women. Additionally, all five forms of exercise significantly improve PWV in this population. Furthermore, INT and HYB significantly improve AIx, while CET, INT, and CT significantly improve NO levels in postmenopausal women. The study did not show a significant reduction in IMT. According to the SUCRA probability ranking, INT and CT are the most effective exercise interventions for improving vascular function in postmenopausal women, while CET and RT are less effective. Therefore, INT and CT are the recommended exercise modalities for improving vascular function in postmenopausal women, while CET and RT are not preferred.

Supplemental Information

Supplemental Information 1 PRISMA checklist.

Supplemental Information 2 Systematic Review and Meta-Analysis Rationale.

Supplemental Information 3 Appendix.

Additional Information and Declarations

Competing Interests

Author Contributions

Data Availability

The authors declare that they have no competing interests.

Weihao Sun conceived and designed the experiments, analyzed the data, prepared figures and/or tables, and approved the final draft.

Yanli Han performed the experiments, analyzed the data, prepared figures and/or tables, and approved the final draft.

Song Gu conceived and designed the experiments, authored or reviewed drafts of the article, and approved the final draft.

The following information was supplied regarding data availability:

The raw data are available at Figshare: Sun, Weihao (2024). original data. figshare. Dataset. https://doi.org/10.6084/m9.figshare.25511797.v1.

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
