# Peer review of "Effects of five types of exercise on vascular function in postmenopausal women: a network meta-analysis and systematic review of 32 randomized controlled trials"

_PeerJ, doi:10.7717/peerj.17621_

## Round 0.1 · original submission · Minor Revisions

Deae authors,

Please make corrections as suggested by the reviewers and provide a detailed rebuttal on a point-by-point basis.

Reviewer 1 ·

Basic reporting

No comment.

Experimental design

No comment.

Validity of the findings

No comment.

Additional comments

The article is very well-written and presented, with a thorough description of the used methodology. The topic is relevant, timely and interesting.

I have a few minor suggestions:

1. consider replacing the term ‘elderly’ with ‘older adult’ or similar, throughout the text, since the term ‘elderly’ is now considered ageist (for reference, please see: https://journals.lww.com/jgpt/fulltext/2011/10000/use_of_the_term__elderly_.1.aspx)

2. I suggest writing again both the full words and their abbreviations (for exercise types and vascular outcome measures) at the first mention in the Introduction section of the article, regardless of the fact that they have already been mentioned in the Abstract (this applies to all abbreviations on their first mention in the manuscript)

3. Tables – please, make sure that all the abbreviations used in each Table are explained in the respective table legend

4. Line 47 – “especially those in the middle-aged and elderly population” – this might even be redundant?

5. Line 141 – Any disagreement was (instead of “shall be”).

Reviewer 2 ·

Basic reporting

no comment

Experimental design

no comment

Validity of the findings

no comment

Additional comments

The way the brackets are arranged in the text sent appears to be slightly inconsistent. Typically, brackets are employed to surround citations or to add details inside of sentences. But it seems like the brackets are positioned erratically in the text submitted.

A brief explanation of why CET and RT are not recommended as first-choice exercises should be considered (including line 32).

It remains speculative in lines 66-67, that Expending at least 1000 kcal of energy through exercise per week can reduce CVD mortality by 20%, while the statement suggests a potential association between a specific exercise regimen and reduced CVD mortality, it lacks sufficient evidence and robust research studies to validate this assertion. Please explain why you used one reference.
In line 131, had "an" intervention duration.

Add an article in line 155 (with f3 domains judged as an unclear risk)

In the literature selection 3.1., it should be mentioned that 168 full-text articles are assessed for eligibility as indicated in Figure 1.

Add an article in line 207 (studies showed a low risk of selective reporting).

Although the research position is made clear, there is no assessment of the strategy's specificity or sensitivity. Including details on the validation or piloting process of the search strategy could improve the study's consistency.

Ultimately, the procedure for gathering the data was methodical and stringent, abiding by the standards and procedures that have been set forth for performing systematic review and network meta-analysis.


The manuscript's methods appear to be thorough and structured, which renders them reproducible to other researchers. The study's methodological robustness is enhanced by the application of well-established guidelines, exacting search approaches, open-ended selection criteria, and comprehensive data synthesis and analysis methodologies.


The information that forms the basis of the conclusions is supplied or made accessible in a reputable discipline-specific repository. The information is controlled, reliable, and sound statistically.

·

Basic reporting

No comment

Experimental design

In line 124, the authors included the study if " (1) it involved postmenopausal or elderly
female participants but again excluded the study (in line 129) if the study " (3) did not involve postmenopausal women." Then why do studies involving the elderly nonmenopausal females were selected?
In line 129, "(2) were non-RCTs" is only the opposite of "(5) it encompassed all English-language RCTs published from 2000 to December 2023" and in line 131, "(6) had intervention duration of less than 6 weeks is opposite to "(4) the experimental group underwent structured exercise training for more than 6 weeks". Please keep only the inclusion criteria for these points.

Validity of the findings

Please add a forest plot.

---

## Round 0.2 · accepted · Accept

Dear Authors, I herewith confirm the acceptance of your paper in its current form.

Reviewer 1 ·

Basic reporting

No further comment.

Experimental design

No further comment.

Validity of the findings

No further comment.

Reviewer 2 ·

Basic reporting

.

Experimental design

.

Validity of the findings

.

Additional comments

The authors carefully addressed the issues that reviewers pointed out, including the abstract, introduction, methods and conclusion. Crucial and carefull changes have been made to the article to add everything that reviewers have commented on. All of the suggestions were addressed in the light shed by the reviews.

·

Basic reporting

No comment

Experimental design

Suggested changes have been made.

Validity of the findings

Suggested changes have been made.